# Analysis of High-Efficiency Mo-Based Solar Selective Absorber by Admittance Locus Method

**Hung-Pin Chen** [1,2] **, Chao-Te Lee** [2]**, Wei-Bo Liao** [1]**, Ya-Chen Chang** [1]**, Yu-Sheng Chen** [1]**, Meng-Chi Li** [3]**, Cheng-Chung Lee** [1] **and Chien-Cheng Kuo** [1,*]

[1] Department of Optics and Photonics/Thin Film Technology Center, National Central University, 300, Chung Da Rd., Chung Li, Taoyuan 32001, Taiwan; chbin@itrc.narl.org.tw (H.-P.C.); w_4240@hotmail.com (W.-B.L.); chen19910130@gmail.com (Y.-C.C.); antiarko@hotmail.com (Y.-S.C.); cclee@dop.ncu.edu.tw (C.-C.L.)

[2] Instrument Technology Research Center, National Applied Research Laboratories, No. 20. R&D Rd. VI, Hsinchu Science Park, Hsinchu 30076, Taiwan; tigerlee@itrc.narl.org.tw

[3] Optical Sciences Center/Thin Film Technology Center, National Central University, 300, Chung Da Rd., Chung Li, Taoyuan 32001, Taiwan; mcli@dop.ncu.edu.tw

* Correspondence: cckuo@ncu.edu.tw; Tel.: +886-3-422-7151-65279

**Abstract:** High absorptivity and low emissivity are characteristics needed in an ideal solar selective absorber. In high-temperature applications, such as a solar concentration power system in which the solar surface works under a long-term high temperature (about 400 to 800 °C), the absorber material has to maintain high absorption in the visible region, high reflectance in the infrared region, and excellent thermal stability at high temperature. In this research, the design of a molybdenum-based (Mo-based) solar selective absorber was analyzed by the admittance locus method, and the films were deposited by magnetron sputtering. The ratio of the extinction coefficient to the refractive index of the Mo layer was close to 1, so that the Mo-based solar selective absorber had a broad absorption band, high absorption, and good solar selectivity. Its average reflectance in the visible region was less than 0.4%. The experimental absorption was 97.1% (simulated absorption was 98%) and the emissivity was from 13% to 20% (simulated emissivity was 8% to 26%) as the temperature increased from 400 to 800 °C.

**Keywords:** solar selective absorber; solar concentration power system; admittance locus method; molybdenum; absorption; emissivity

## 1. Introduction

Solar energy is a clean source of renewable energy and it is also the most abundant. Photon–electrical and photon–thermal conversions are the two most practical versions of solar energy application. The photon–electrical conversion solar cell can be easily applied in many regions, but it has some disadvantages, such as a narrow absorption spectrum, limited available material, and high production cost. In contrast, the photon–thermal conversion system is low-cost, and it can easily realize high absorption and low thermal emittance through structural optimization. The concentrated solar power (CSP) system is one of the most common photon–thermal conversion systems. The CSP system uses a solar selective absorber to convert sunlight to solar thermal energy. The system is heated at over 400 °C; therefore, it is a challenge for the solar selective absorber to achieve high absorption, low thermal emissivity, and good thermal stability under high-temperature conditions.

In the last few decades, considerable efforts have been channeled toward developing high and broadband solar selective absorbers for photon–thermal conversion [1]. The ideal solar selective absorber absorbs sunlight by trapping the photons in the layered structures and reflects

far-infrared light to prevent thermal radiative emission from the substrate [2]. The optical behavior strongly depends on the optical performance of solar selective materials and the layered structure. To obtain outstanding spectral selectivity, various approaches have been investigated, such as semiconductor-metal tandems [3–5], metal-dielectric composite coatings [6–8], textured surfaces, and optical interference multilayer surfaces [9–12]. Zhou et al. indicated that a metal-dielectric structure had a good spectral selective characteristic and thermal stability [5]. Xiao et al. deposited $Al_2O_3$:Ag thin films by magnetron sputtering [6]. When the thickness was larger than 120 nm, the absorption was almost unchanged even with annealing at 700 °C for 64 h. Although the thermal stability was good, the absorptivity in the 0.3–2.5 μm region was only 91% [6]. Some researchers have made efforts to fabricate nanostructures, such as 1D gratings [13], 2D gratings [14], and 3D photonic crystals [15]. Nanostructured optical interference films are easier to manufacture, and they have good spectral selectivity, excellent thermal stability, operate under high-temperature conditions, and absorb solar energy effectively. However, the optimal design of a metal-dielectric structure has not been determined using the optical interference theorem [6–8]. The admittance locus method is a kind of optical interference films design technique that has since been used in the modeling and design of filters, and it has been extensively used to predict the performance of optical interference films.

In this study, we analyzed the layered structure of a solar selective absorber with different kinds of reflection metal layers and absorption metal layers based on the admittance locus method. The results showed that the molybdenum-based (Mo-based) solar selective absorber provided high photo-conversion efficiency and excellent thermal stability.

## 2. Theory and Simulation

A high-efficiency solar selective absorber has high absorption in the visible and near-infrared regions and low radiation in the wavelength range greater than the middle-infrared region, as shown in Figure 1. The cut-off wavelength ($\lambda_c$) is dependent on the application temperature. The reflectance at the wavelength less than the cut-off wavelength will be low, while that at the wavelength greater than the cut-off wavelength will be high. According to Planck's law, the radiation intensity of blackbody ($E_{b\lambda}$) is a function of wavelength ($\lambda$) and temperature ($T$) as follows:

$$E_{b\lambda} = \frac{C_1 \lambda^{-5}}{e^{C_2/\lambda T} - 1} \tag{1}$$

where $C_1$ and $C_2$ are $3.742 \times 10^{-16}$ W·m$^2$ and $1.438 \times 10^{-2}$ m·k, which are Planck's first and second radiation constants, respectively. According to Kirchhoff's law, the radiation in the wavelength range from $\lambda_1$ to $\lambda_2$ is described as:

$$F_b(\lambda_1 \sim \lambda_2) = \frac{\int_{\lambda_1}^{\lambda_2} E_{b\lambda} d\lambda}{\int_0^\infty E_{b\lambda} d\lambda} = \frac{\int_{\lambda_1}^{\lambda_2} E_{b\lambda} d\lambda}{\sigma T^4} \tag{2}$$

where $\sigma T^4$ is adopted from the Stefan–Boltzmann law.

Thus, $F_b(0 \sim \lambda)$ can be defined as:

$$F_b(0 \sim \lambda) = \frac{\int_0^\lambda E_{b\lambda} d\lambda}{\sigma T^4} \tag{3}$$

To simplify the calculation, we assume that the solar selective absorber has a reflectance of 0.9 at wavelengths above the cut-off wavelength and 0.1 at wavelengths below the cut-off wavelength. Then, using Equation (4) and approximating the sun blackbody radiation temperature as 6000 K, the absorption of the solar selective absorber is as given in Equation (4) because the transmittance is 0.

$$\alpha = 1 - R = 0.9 \times F_b + 0.1 \times (1 - F_b). \tag{4}$$

Kirchhoff's law shows that the absorption ($\alpha$) is equal to the emissivity ($\varepsilon$); thus, it is easy to calculate the thermal emissivity ($\varepsilon$) of the film at 673 K and obtain the solar selectivity $\alpha/\varepsilon$ at different cut-off wavelengths as shown in Figure 2.

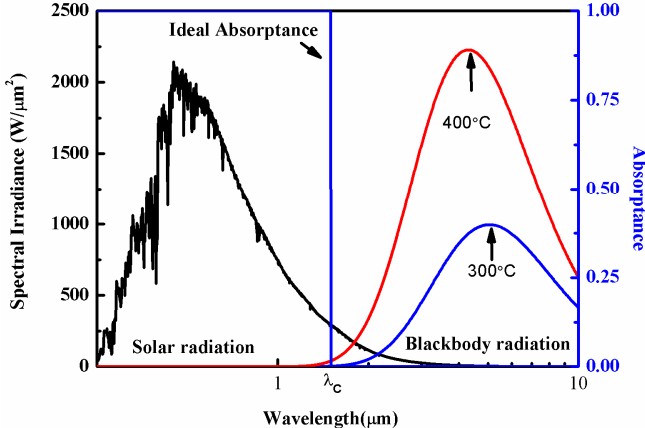

**Figure 1.** Solar radiation and blackbody radiation spectra.

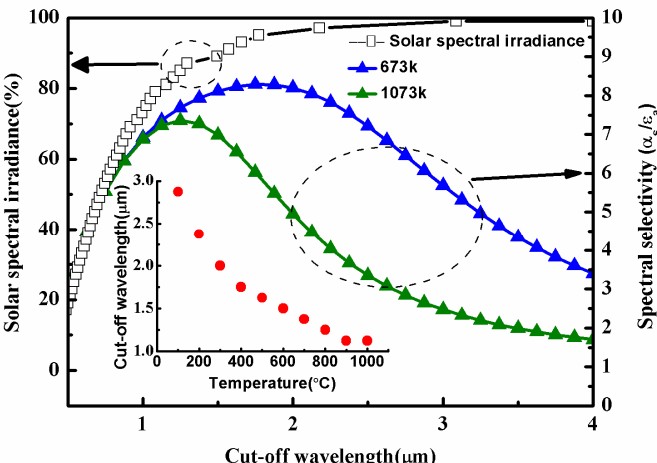

**Figure 2.** Change of solar selectivity $\alpha/\varepsilon$ at 673 and 1073 K with the cut-off wavelength; the optimal cut-off wavelength at different temperatures of 100–1000 °C.

The peak of spectral selectivity in Figure 2 is the result of evaluating the optimal cut-off wavelength for the solar selective absorber for different temperatures. The operating temperature for the CSP system is 400–800 °C; therefore, the cut-off wavelength is 1.5 μm for our design.

The solar selective absorber with a metal-dielectric structure has a high broad absorption in the visible and near-infrared regions and reflects far infrared to reduce the thermal radiation loss from the substrate. For energy conservation, the sum of transmittance, reflectance, and absorptivity must be satisfied. Since the thickness of the reflection metal layer is thick enough, the transmittance of the solar selective absorber is 0. The absorptivity has to be deduced from the calculation of reflectance.

The admittance locus method is a well-known method to simulate optical thin film multilayer structures. The multilayer coating is gradually grown on the substrate layer by layer and immersed all the time in the final incident medium. When the thickness of each layer increases from 0 to its final value, the admittance is calculated and the locus is plotted. The admittance locus includes the phase thickness and the equivalent admittance (optical constant) of the deposited film during the film deposition process. If the refractive index and physical thickness can be obtained from the transmittance or reflectance spectrum in advance, the admittance locus of a depositing film can be predicted. Considering a beam of light is normally incident to the depositing film, to satisfy

the electromagnetic boundary conditions at the interfaces in the multilayer structure, the following equation should be satisfied:

$$
\begin{bmatrix} B \\ C \end{bmatrix} = \prod_{j=1}^{z} \begin{bmatrix} \cos \delta_j & \frac{i}{y_j} \sin \delta_j \\ i y_j \sin \delta_j & \cos \delta_j \end{bmatrix} \begin{bmatrix} 1 \\ y_s \end{bmatrix} \tag{5}
$$

where $y_j$ and $\delta_j$ represent the admittance and phase thickness in the $j$th medium, respectively, and $\delta_j = 2\pi n_j d_j / \lambda_j$, where $n_j$, $d_j$, and $\lambda_j$ are the refractive index, physical thickness, and reference wavelength of the $j$th layer, respectively. If the multilayer structure is deposited, its equivalent admittance will be $y_E = H/E = C/B$, where $H$ and $E$ are magnetic and electric fields, respectively. The corresponding transmittance ($T$) and reflectance ($R$) of the multilayer are as follows:

$$
T = \frac{4 n_0 y_s}{(n_0 B + C)(n_0 B + C)^*}; \ R = \left( \frac{n_0 B - C}{n_0 B + C} \right) \left( \frac{n_0 B - C}{n_0 B + C} \right)^* \tag{6}
$$

where $n_0$ is the incident medium refractive index.

The reflection layer has to reflect infrared light to avoid thermal radiation from the substrate. Metals such as Ag, Al, Cu, and Mo have high reflectivity in the infrared region, as shown in Figure 3. These metals can be chosen as candidates for the reflection layer. It is worth noting that Mo has a lower reflectivity (i.e., higher absorptivity) in visible and near-infrared regions than the other metals. For operations in high temperature, the thermal stability of the metals needs to be considered. Table 1 presents the melting points of the metals, and Mo is shown to have the highest melting point. Lin et al. analyzed the stability of ultrathin Mo nanowire by molecular dynamics simulation [16]. The results indicated that the melting points were over 1230 K. Although the melting point was lower than the bulk value, the ultrathin Mo could still be applied in solar selective absorbers and other heat-resistant devices. Molybdenum is the most suitable material for reflection layer.

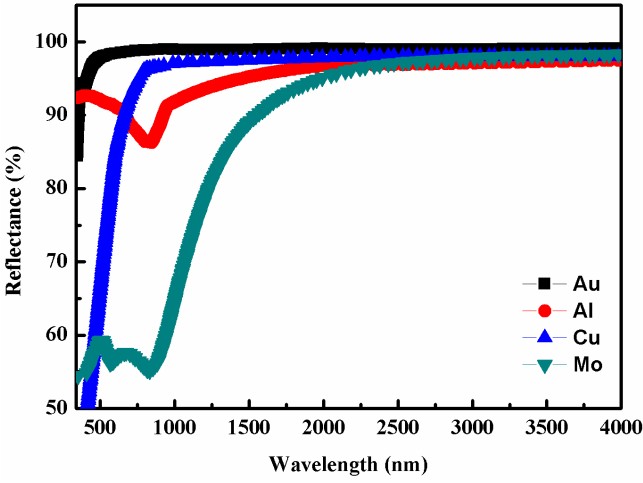

**Figure 3.** The reflectance spectrum of simulation metal films with a thickness of 150 nm.

**Table 1.** The melting point of metals.

| Material | Ag | Al | Cu | Mo |
|---|---|---|---|---|
| Melting point (°C) | 961 | 660 | 1084 | 2623 |

The absorptivity of the solar selective absorber strongly depends on the absorption layer. Figure 4 shows different absorption layers on a B270/Mo/Nb$_2$O$_5$ structure, and distinct admittance loci can be recognized easily due to different optical constants. The admittance locus of silver (Ag) thin film decreased rapidly. In contrast, the admittance loci variation of Mo and chromium (Cr) followed an

almost horizontal path. The differences were caused by the ratio of the extinction coefficient (*k*) to the refractive index (*n*). When the *k*/*n* ratio was large, the imaginary part of admittance reduced more quickly than the real part, resulting in a path like the admittance locus of Ag.

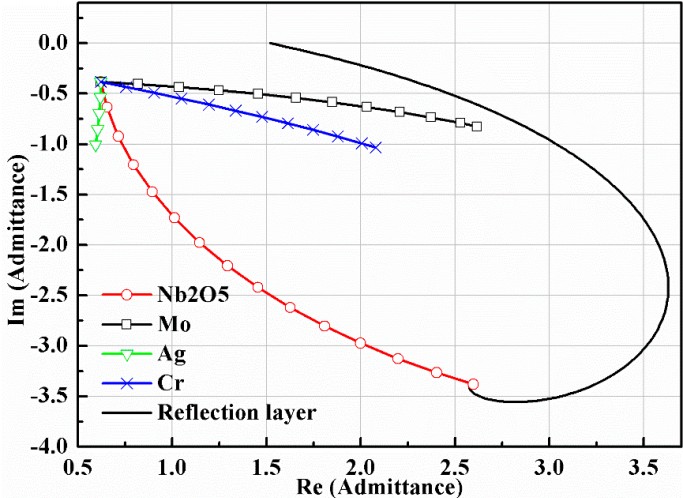

**Figure 4.** Admittance loci for different absorption layers.

The admittance loci of (B270/Mo/$Nb_2O_5$/absorption layer/$Nb_2O_5$/$SiO_2$) designed to absorb solar radiation from the visible region for three wavelengths of 400, 550 and 700 nm are illustrated in Figure 5. The loci of 400 nm (Figure 5a) were very different from those of 550 nm and 700 nm. The final admittance value was very different because of the mismatched optical constants of Ag. To increase absorption in the visible region, the final value of admittance from 400 to 700 nm must be as close as possible to 1 (refractive index of air). As seen in Figure 5a,b, the admittance loci for wavelengths of 400, 550 and 700 nm were largely different; therefore, the absorption regions were narrow owing to the mismatch of the optical constants of Ag and Cr thin film. However, the admittance loci of Mo thin film in the three wavelengths were close, as shown in Figure 5c. The absorption region of 400 to 700 nm was broad.

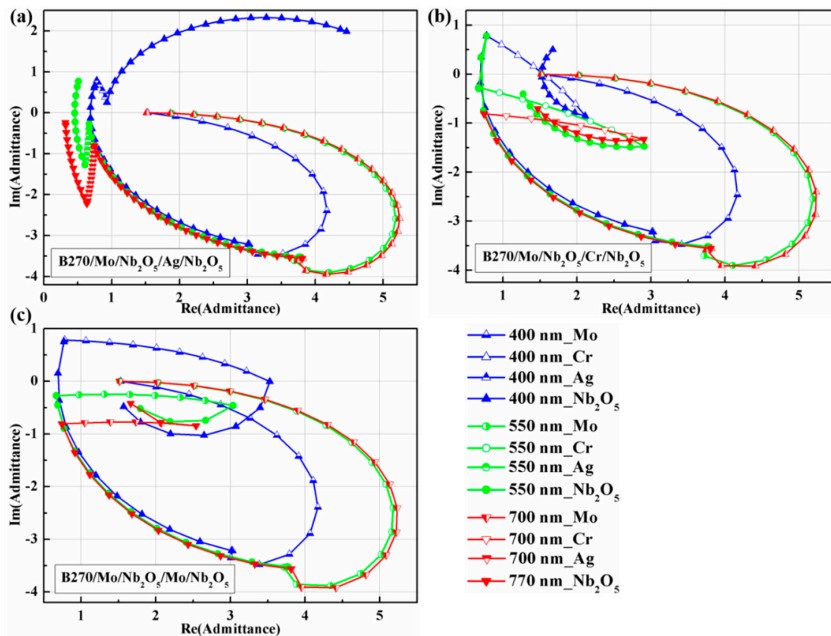

**Figure 5.** Admittance loci of solar selective absorber using (**a**) Ag, (**b**) Cr, and (**c**) Mo as the absorption layer for wavelengths of 400 nm, 550 nm, and 700 nm.

### 3. Experimental

Films were deposited on a B270 glass substrate (2.54 cm × 2.54 cm) via a direct current (DC) magnetron sputtering system using 3 inch molybdenum (Mo) and 6 inch silicon (Si) and niobium (Nb) as the sputtering target. Figure 6 shows the schematic diagram of the magnetron sputtering system. The distance from the target to the substrate was 80 mm. During deposition, the base pressure was lower than $8 \times 10^{-6}$ torr and was maintained by a cryopump, and the working gases were argon and oxygen. The transmittances in the visible and near-infrared regions were measured using a Hitachi U4100 spectrometer (Tokyo, Japan), and the reflectance in the infrared region was measured using a Fourier-transform infrared spectrometer with a 30° specular reflectance in the range of 400 to 4000 $cm^{-1}$. The optical constant of Mo thin films was analyzed using an ellipsometer (J. A. Woollam, Lincoln, NE, USA) and a Drude model. Figure 7 shows the optical constant of the Mo reflective layer when the thickness was 150 nm. The refractive index and extinction coefficient of the Mo absorption layer for different deposition powers are shown in Figures 8 and 9, respectively. The optical constants of the Mo absorption layer changed quickly for different deposition powers because the thickness was only about 7.16 nm. Under higher deposition powers, the optical constants of the Mo absorption layer were more similar than those of the Mo reflective layer. In this study, the deposition power for the Mo absorption layer was 400 W.

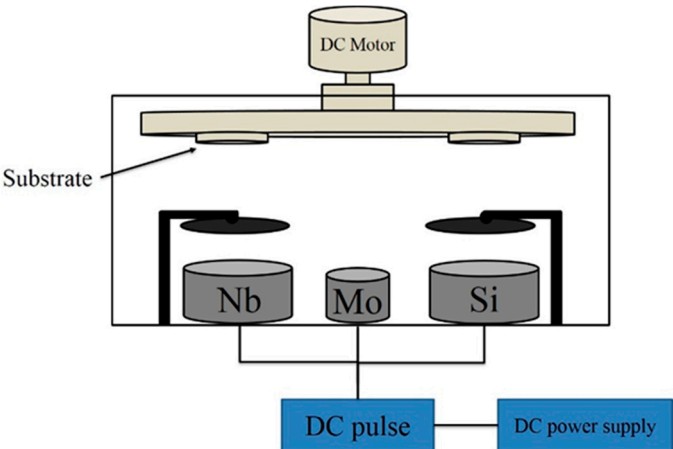

**Figure 6.** Schematic diagram of the DC magnetron sputtering system.

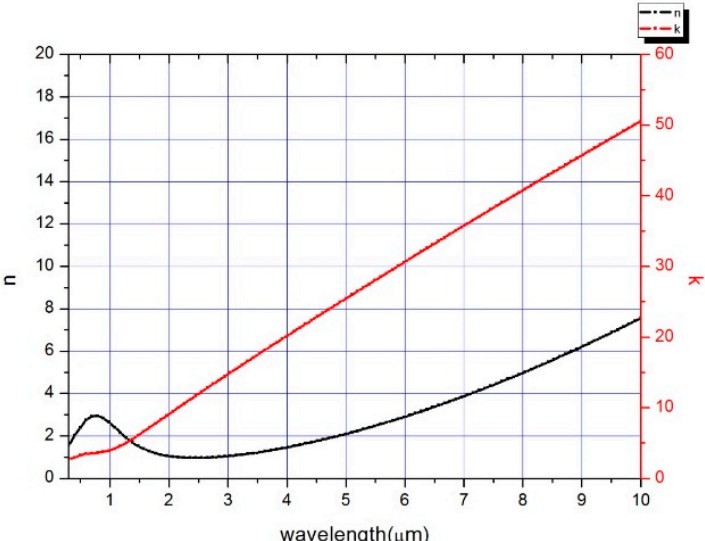

**Figure 7.** The optical constant of the Mo reflective layer.

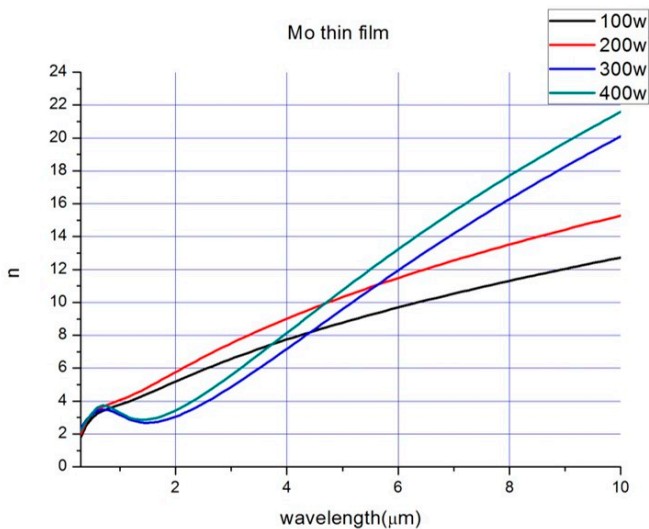

**Figure 8.** The refractive index of the Mo absorption layer for different deposition powers.

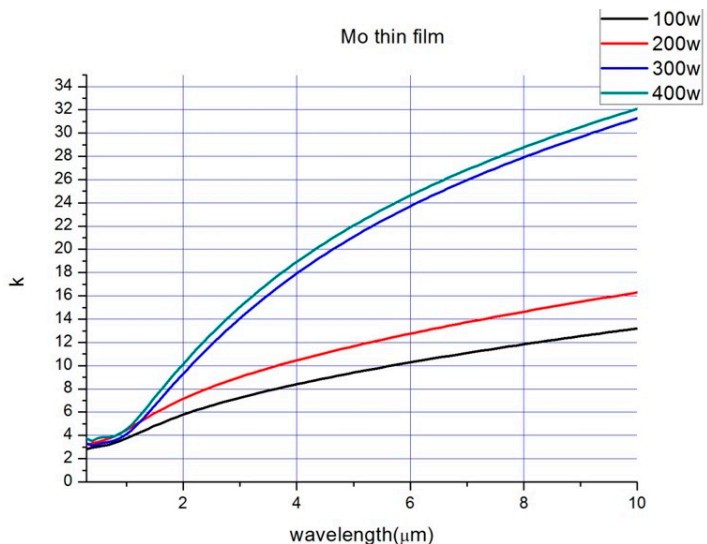

**Figure 9.** The extinction coefficient of the Mo absorption layer for different deposition powers.

## 4. Results and Discussion

The optical constants of $SiO_2$, $Nb_2O_5$, Mo reflective layers, and of Mo absorption layer at 550 nm were 1.47, 2.32, 2.59–3.38i, and 3.44–3.82i, respectively. Figure 10 shows the optimization design of the solar selective absorber, and the admittance loci at 550 nm are shown in Figure 11. First, the Mo reflective layer with a thickness of 150 nm was deposited on B270; then, the admittance loci transformed from $Y_s = 1.52$ to $Y_m = 2.59–3.38i$. The admittance loci of the final structure were located in $Y_4 = 1.15–0.06i$. Moreover, $Y_4$ was close to 1 ($Y_{air}$) and attained an anti-reflection property. The key point was the second metal layer (absorbing layer). When the $k/n$ ratio of the Mo absorption layer was close to 1, the admittance locus $Y_2$ shifted horizontally.

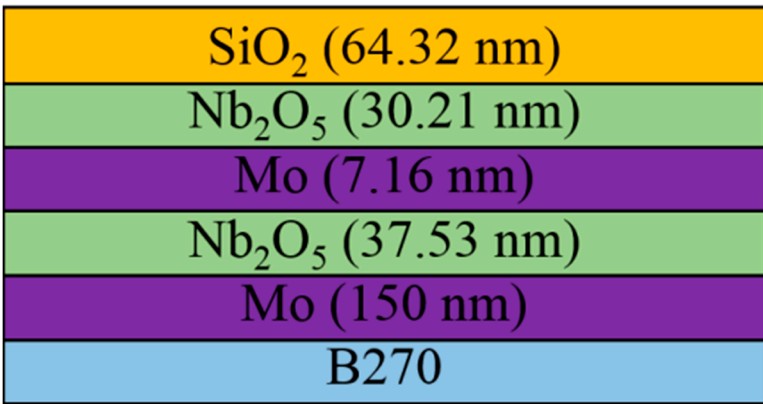

**Figure 10.** Optimization design of solar selective absorber.

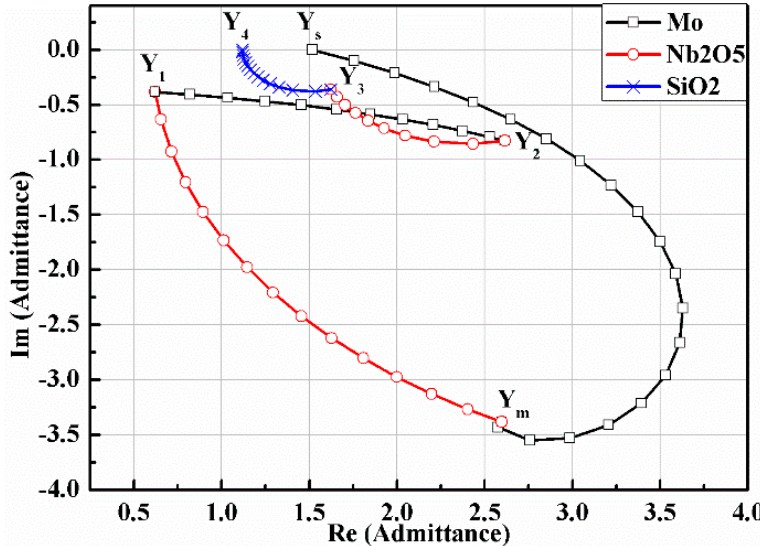

**Figure 11.** Admittance loci at 550 nm of optimization solar selective absorber.

The reflection spectra of the fabrication and simulation are shown in Figure 12. The reflectance spectrum of the simulation in the visible region was less than 1%. The average reflectance in the visible region was less than 0.4%. Although the reflectance of the fabrication was higher than that of the simulation in the wavelength range below 500 nm, the sample had good absorption (97%) in the visible region.

According to Kirchhoff's law, the emissivity of a heat-radiated body or surface at thermal equilibrium is a function of the reflectance $R$, as shown in Equation (7). The calculated results are shown in Table 2. The simulated absorption and emissivity at 500 °C were over 98% and 11%, respectively. The absorption and emissivity of fabrication at 500 °C were 97% and 15%, respectively. The absorption was 2% higher than that obtained by Zhou et al. [3]. The difference is due to the absorption layer. The thickness of the Mo thin film was only about 7 nm. It is difficult to control a metal thin film when its thickness is less than 10 nm.

$$\alpha(\theta, T) = \frac{\int_0^\infty [1 - R(\theta, \lambda)] dE_{b\lambda}(\lambda, T)}{\int_0^\infty dE_{b\lambda}(\lambda, T)} \tag{7}$$

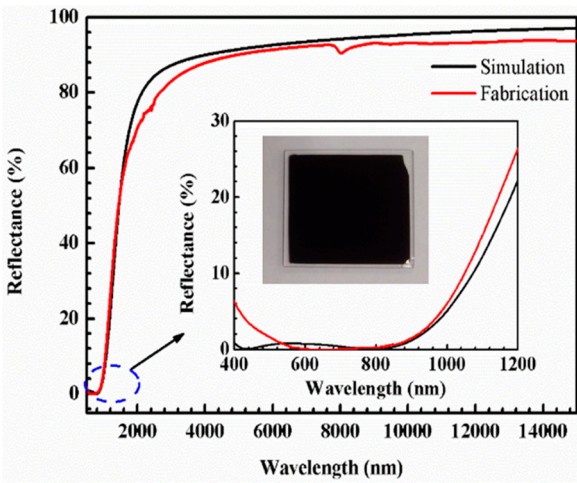

**Figure 12.** Reflectance spectra of the simulation and fabrication.

**Table 2.** Calculated absorption and emissivity of a solar selective absorber.

| Temperature | Simulation | | | Fabrication | | |
|---|---|---|---|---|---|---|
| | Absorption ($\alpha$) | Emissivity ($\varepsilon$) | Solar Selectivity ($\alpha/\varepsilon$) | Absorption ($\alpha$) | Emissivity ($\varepsilon$) | Solar Selectivity ($\alpha/\varepsilon$) |
| 400 °C | | 0.08 | 12.26 | | 0.13 | 7.46 |
| 500 °C | | 0.11 | 8.92 | | 0.15 | 6.47 |
| 600 °C | 0.9814 | 0.15 | 6.54 | 0.9703 | 0.16 | 6.06 |
| 700 °C | | 0.2 | 4.9 | | 0.18 | 5.39 |
| 800 °C | | 0.26 | 3.77 | | 0.2 | 4.8 |

## 5. Conclusions

In this study, the structure of a solar selective absorber was optimized using the admittance locus method. From the analysis, the key point of a high-efficiency solar selective absorber was found to be the k/n ratio of the absorption layer. When the k/n ratio of the Mo absorption layer was close to 1, the admittance locus shifted horizontally. The absorption region band was broad and the absorption was high. The simulated absorption in the solar radiation region was 98%. The simulated emissivity ranged from 8% to 26% with increasing the temperature from 400 to 800 °C. The experimental absorption was 97.1%, which was 2% higher than that in the published research. The experimental emissivity at temperatures of 400 to 800 °C ranged from 13% to 20%.

**Author Contributions:** Conceptualization, C.-C.K. and C.-C.L.; Methodology, H.-P.C. and C.-T.L.; Software, W.-B.L. and Y.-S.C.; Validation, W.-B.L., Y.-C.C. and Y.-S.C.; Formal Analysis, W.-B.L. and Y.-C.C.; Investigation, M.-C.L.; Resources, H.-P.C.; Data Curation, C.-T.L.; Writing—Original Draft Preparation, Y.-S.C.; Writing—Review and Editing, W.-B.L. and Y.-C.C.; Visualization, C.-C.K.; Supervision, C.-C.K.; Project Administration, H.-P.C.; Funding Acquisition, C.-C.K. and C.-C.L.

**Funding:** This research was funded by the Ministry of Science and Technology (Nos. MOST 107-2622-E-008-013 -CC2 and MOST 107-2221-E-008-067).

**Conflicts of Interest:** The authors declare no conflict of interest.

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
