# Peer review of "Analysis of High-Efficiency Mo-Based Solar Selective Absorber by Admittance Locus Method"

_coatings, doi:10.3390/coatings9040256_

Round 1
Reviewer 1 Report
The manuscript can be published with minor correction in the introduction section.
(1) Add few important references In line 40, after ...photo-thermal conversion.
(2) Add one reference at the end of line 42 after ... ... from substrate.
(3) In line 45, you have included all 12 references in place. I suggest you to divide these references for specific cases. For example, add some of these references after semiconductor-metal tandems, some after metal-dielectric compositing coatings, some after textured surface and optical interference multilayer.
(4) Sentence in line 59-60 is not complete.
(5) Please check English language thoroughly.
Author Response
Thanks for your reviewing
(1) Add few important references In line 40, after ...photo-thermal conversion.
Add reference in line 41
(2) Add one reference at the end of line 42 after ... ... from substrate.
Add reference in line 43
(3) In line 45, you have included all 12 references in place. I suggest you to divide these references for specific cases. For example, add some of these references after semiconductor-metal tandems, some after metal-dielectric compositing coatings, some after textured surface and optical interference multilayer.
Divide these references in line 41 to 47.
(4) Sentence in line 59-60 is not complete.
Correct the sentence in line 64 to 66
(5) Please check English language thoroughly.
Complete English language editing
Reviewer 2 Report
I cannot find any progress or further improvement compared to previous reports about many different types of Mo-solar cell. The novelty of the current work is rather weak. The manuscript needs to be severely polished in order to become acceptable.
1. The paper is badly written, has some grammatical mistakes, such as “ The solar energy is the most abundant source of renewable energy and play an important part” , and “ellipseometer”. And the English should be polished.
2. The format and punctuation of the article must be standardized.
3. Could the authors provide the ellipsometer model and the optical costants (n and k from the Drude Model ( line 170) ? Moreover, if it is possible, SEM images of the Mo thin film films could be useful.
4. The authors should discuss the novelty of their work in comparison to previous works.
I would suggest the authors to rewrite the Introduction section to make the aim of their work clearer.
Author Response
Thanks for your reviewing
1. The paper is badly written, has some grammatical mistakes, such as “ The solar energy is the most abundant source of renewable energy and play an important part” , and “ellipseometer”. And the English should be polished.
Complete English language editing
2. The format and punctuation of the article must be standardized.
Correct the format and punctuation
3. Could the authors provide the ellipsometer model and the optical costants (n and k from the Drude Model ( line 170) ? Moreover, if it is possible, SEM images of the Mo thin film films could be useful.
Add description in line 162 to 168, and figure 7-9 to explain the optical constants of Mo reflective layer and Mo absorption layer.
4. The authors should discuss the novelty of their work in comparison to previous works.
I would suggest the authors to rewrite the Introduction section to make the aim of their work clearer.
Add description in line 53 to 62 to explain the aim of this manuscript
Round 2
Reviewer 2 Report
The authors have nicely addressed my concerns. I suggest the authors to change the spelling of "ellipseometer" .The correct form is "ellipsometer".